# Advances in Beet (*Beta vulgaris* L.) Stress Adaptation: Focus on Transcription Factors and Major Stress-Related Genes

**DOI:** 10.3390/plants15010012

**Published:** 2025-12-19

**Authors:** Guan Liu, Yifei Tang, Hanhui Wang, Song Yu, Huan Gao, Yang Wang, Dongye Zhang

**Affiliations:** 1College of Advanced Agriculture and Ecological Environment, Heilongjiang University, Harbin 150080, China; 2019057@hlju.edu.cn (G.L.);; 2State Key Laboratory of Tree Genetics and Breeding, College of Forestry, Northeast Forestry University, Harbin 150040, China

**Keywords:** Beet (*Beta vulgaris* L.), abiotic/biotic stresses, stress tolerance, key genes, regulation

## Abstract

Beet (*Beta vulgaris* L.) is an important economic crop widely cultivated across various regions worldwide. Its agricultural significance lies not only in its high sugar yield but also in its positive impact on agro-ecosystems and the economic value of its by-products. However, beet production and quality are adversely affected by multiple abiotic and biotic stresses, including pathogen infection, drought, salinity, and extreme temperatures. In recent years, numerous key stress-responsive genes have been identified, including *BvPAL*, *BvPR*, and *Rz1-4*, which mediate responses to biotic stresses, and *BvM14-SAMS2*, *BvINT1;1*, *BvHMA3*, *BvCOLD1*, and *BvALKBH10B*, which enhance tolerance to abiotic stresses. Meanwhile, core transcription factors such as bHLH, HSP, WRKY, and SPL show differential expression under stresses, suggesting that they may regulate stress-related genes and constitute major transcriptional modules enabling beet to withstand adverse conditions. In this study, we summarize the changes in beet under different stress conditions, combining gene information to reveal key regulatory changes in stress responses and how these molecular processes contribute to stress adaptation. This not only provides a theoretical basis for the improvement of beet stress tolerance and yield, but also offers potential directions for future breeding strategies in practical applications.

## 1. Introduction

Beet (*Beta vulgaris* L.), belongs to the family Amaranthaceae, is an agriculturally important crop and the second-largest global source of sugar after sugarcane (*Saccharum officinarum* L.) [1]. As a key root crop, its main root is used in the sugar production process, not only serving as a source of sucrose but also being utilized in the production of bioethanol, biodegradable polymers, and biofertilizers [2,3,4]. Beet originated from *Beta vulgaris* L. ssp. *maritima* (sea beet or wild beet) native to the Mediterranean coastal regions [5,6,7]. Through a long evolutionary process, it developed strong tolerance to cold, drought and salt, allowing it to thrive across areas such as the Eastern European plains and Northwestern Europe [5]. Beet’s adaptability arises from its complex interactions with the environment, allowing it to respond to various biotic and abiotic stresses through the regulation of physiological and molecular processes. However, climate change and pathogen spread have intensified these stresses, leading to considerable yield losses [8]. Thus, elucidating the mechanisms underlying beet–environment interactions is essential for improving stress tolerance and achieving sustainable production.

In plant–environment interactions, stresses are mainly categorized as biotic, such as pathogen infection and insect attack, and abiotic, including salinity, drought, and extreme temperatures [9]. Among the various stresses affecting beet, biotic stresses are primarily caused by pathogens and pests, including fungi (e.g., *Cercospora beticola*), viruses (e.g., beet necrotic yellow vein virus), bacteria (e.g., *Pseudomonas syringae* pv. *aptata*), and insect pests (e.g., *Spodoptera litura*) [10,11,12,13]. These biotic stresses could damage leaf and root tissues, directly reducing photosynthetic efficiency, yield, and sugar accumulation. Moreover, during beet growth, abiotic stresses such as salinity, drought, high temperature, and heavy metal exposure could disrupt cellular homeostasis, thereby impairing normal plant development [14,15,16,17,18,19]. Notably, frequent stress events and their interactions—such as the combined effects of drought and high temperature—continue to pose significant challenges to beet production, underscoring the urgent need to elucidate the underlying molecular response mechanisms.

Stress signaling and the associated molecular responses are central to plant development in response to environmental challenges. Upon stress exposure, beet could perceive external cues and subsequently activate signaling networks through complex signal sensing and molecular response mechanisms [20,21]. For example, under abiotic stress, the abscisic acid (ABA) signaling pathway is rapidly activated, and SnRK2 kinases could phosphorylate downstream transcription factors to trigger the expression of defense genes associated with ion transport and osmotic regulation, thereby mitigating ion toxicity and water imbalance [22]. While under biotic stress, such as pathogen attack, jasmonic acid (JA), ethylene (ET), and salicylic acid (SA) signaling pathways function synergistically: SA activates the expression levels of pathogenesis-related gene via *NPR1*-mediated systemic acquired resistance (SAR), primarily targeting biotrophic pathogens, whereas JA and ET act through the *COI1–JAZ* and *EIN3/EIL* modules to defend against necrotrophic pathogens [23]. Crosstalk occurs among these signaling pathways—for instance, SA could suppress JA responses, and ABA and SA also interact in stress integration [24,25]—although the precise mechanisms in beet remain unclear. In plant stress responses, specific transcription factors, including bHLHs, HSPs, WRKYs, and SPLs, play central regulatory roles by modulating gene expression in response to stress signals, thereby maintaining cellular homeostasis. For instance, under salt and drought stress, bHLHs could regulate *SnRK* genes via ABA signaling, leading to reduced cellular electrolyte leakage and enhanced osmotic adjustment [26]. While under heat and drought stress, HSPs could integrate SA, JA, and ABA signaling to modulate antioxidant defenses, secondary metabolism, and developmental plasticity [27]. Other transcription factors, including NAC, MYB, WRKY, and SPL, also show pronounced differential expression under various stress conditions, indicating their role in coordinating plant stress responses [28]. These transcription factors bind to the promoters of their targets to regulate a suite of stress-responsive genes; however, their specific targets and functional diversity in beet remain to be fully elucidated.

Additionally, secondary metabolism is essential for stress tolerance, particularly in reducing oxidative damage and enhancing plant defenses. Betanin, a prominent compound in beet, not only serves as a natural red pigment widely used in the food and pharmaceutical industries, but also exhibits strong antioxidant and free radical-scavenging activities, while functioning as an osmoprotectant to help the plant cope with adverse environmental conditions [29]. Under salt, drought, stress or pathogen infection, the accumulation of betanin and other secondary metabolites in beet helps alleviate oxidative damage, protects cellular membranes and proteins from reactive oxygen species (ROS) [30]. However, the potential role of betanin in regulating signaling networks to enhance stress defenses remains to be further investigated.

Although the sequencing of the beet genome has laid a foundation for investigating stress-resistance mechanisms [31], and multi-omics approaches have revealed numerous key genes, substantial knowledge gaps remain [32,33]. Beet’s complex genetic background and the absence of a stable transformation system, coupled with studies mostly restricted to laboratory conditions, have limited detailed investigations into its stress signaling and molecular network mechanisms. Therefore, this review aimed to summarize and analyze the differences in abiotic and biotic stress tolerance among different beet varieties, particularly their growth performance in response to stresses such as pathogen infection, drought, salinity, and extreme temperatures. Particularly, we provided an overview of transcription factors and the core genes associated with these stresses and explore their roles in enhancing beet’s stress resistance. In this study, we aim to integrate the scattered gene information in beet to elucidate the key regulatory changes involved in stress responses, to summarize how these molecular mechanisms collectively contribute to stress response. This not only provides a theoretical foundation for a deeper understanding of beet’s stress tolerance mechanisms, but also offers valuable insights for future molecular breeding efforts aimed at improving beet’s stress resistance and increasing yield.

## 2. The Biotic Stress Responses of Beets

### 2.1. Fungal Pathogenesis of Cercospora Leaf Spot Caused by Cercospora beticola

Cercospora leaf spot (CLS), caused by *Cercospora beticola*, is the most destructive foliar disease in sugar beet, and resistant varieties could delay its onset, reducing symptom severity and spore dispersal [34,35]. The mechanisms of beet defense against CLS include key sites and genes such as *phenylalanine ammonia-lyase* (*BvPAL*), *cinnamic acid 4-hydroxylase* (*BvC4H*), *multiprotein bridging factor 1c* (*BvMBF1c*) and *BvbZIPs*, hormone signaling including ABA and JA, antimicrobial enzymes and secondary metabolites.

Studies have shown that CLS resistance is mainly controlled by 4–5 quantitative trait loci (QTLs), with two particularly strong QTLs contributing significantly to resistance [36,37]. Additionally, ISSR, SRAP, and RAPD marker analyses revealed genetic polymorphisms among 8 beet varieties, and validated the effectiveness of enzymes such as peroxidase, polyphenol oxidase, and chitinase in distinguishing resistant and susceptible varieties [38] (Table 1). At the molecular level, beet defense involves multiple regulatory factors. In the early stages of *C. beticola* infection, the expression of *BvPAL* and *BvC4H* was reduced, with *BvPAL* showing a reduction at both the transcriptional and enzymatic levels [39]. ABA concentration in beet leaves initially decreased and then increased, reaching levels similar to those of drought-stressed plants after 15 days [40]. The promoter–luciferase reporter system further revealed the spatiotemporal dynamics of ABA in beet during *C. beticola* infection. The ABA-responsive promoter containing the *AREB cis*-element A2 was locally activated at infection sites, with its activity closely matching measured ABA levels, while *BvAREB1* expression was induced under both drought stress and fungal infection. Additionally, as a marker of defense gene activation, the activity of the beet *BvPAL* promoter was markedly reduced upon ABA treatment. These findings indicated that ABA accumulation and its signaling pathway were likely the main factors responsible for suppressing *BvPAL* expression (Table 1). Another gene, *BvMBF1c*, showed pathogen-responsive expression peaking at 48 h post-inoculation with *C. beticola* and formed a novel interaction with Bvtrehalose-6-phosphate synthase (BvTPS5), which could regulate osmotic balance through trehalose biosynthesis [41]. Moreover, six genes in the BvbZIP family are significantly differentially expressed during infection, with BVRB_9g222570 (encodes a PR1 protein) being associated with SAR [42] (Table 1).

Hormone dynamics analysis revealed that both resistant lines (e.g., 81GM241, F85621) and susceptible lines (e.g., KWS6661, KWS4502, KWS5145) of beet underwent significant changes in endogenous hormone levels following *C. beticola* infection. For instance, in the resistant beet line 81GM241, indole-3-acetic acid (IAA) levels declined during the early-to-mid stages of infection, whereas they increased in the susceptible line KWS4502 [41]. In resistant lines (81GM241 and F85621), endogenous JA and SA levels rose rapidly following infection, while hormone accumulation in susceptible lines (KWS6661 and KWS5145) lagged behind [42,43]. Overall, resistant lines exhibited faster and more stable hormonal responses, promoting early defense activation, whereas susceptible lines showed delayed and unstable hormone signaling, which may contribute to rapid disease progression.

Transcriptomic analysis further compared the defense strategies of resistant (81GM241) and susceptible (KWS6661) varieties across four infection stages (0, 10, 20, and 30 days post-inoculation) [44]. The resistant variety adopted a three-phase defense strategy: in the early stage (10 dpi), reactive oxygen species were scavenged by glutathione-related enzymes, cell walls were reinforced through remodeling by pectin methylesterase, and lignin derived from phenylpropanoid metabolism strengthens structural defenses; during the mid-stage (20 dpi), JA-mediated defenses were amplified, PR1 receptors activated MAPK-WRKY signaling, and flavonoid antimicrobial compounds were mobilized to suppress the pathogen; in the late stage (30 dpi), photosynthetic resources were reallocated to maintain defense via tryptophan-derived antimicrobial secondary metabolites without compromising growth. In contrast, the susceptible variety showed delayed ROS detoxification and impaired signal transduction, ultimately leading to metabolic collapse [44]. These findings all indicate that beet defense against CLS is a multilayered and dynamically regulated process (Table 1).

Moreover, root rot (RR) in beet, caused by *Fusarium* spp., was countered by two key chitinase-encoding genes, *BvSP2* and *BvSE2*. They likely act by hydrolyzing chitin, a key component of the fungal cell wall, thereby disrupting *F. oxysporum*, inhibiting its growth and infection, and enhancing beet resistance to RR. Their responses were both strain-specific and time-dependent: *BvSP2* responded rapidly to the laboratory-inoculated strain No. 5 (*F. oxysporum* Sch.), with resistant lines peaking at 14 days post-inoculation, whereas susceptible lines showed delayed expression, reducing resistance. In contrast, *BvSE2* primarily responded to other naturally occurring field strains, with strong expression detected only in the leaves of susceptible lines [45].

In summary, beet mounts a dynamic, multilayered defense against CLS and RR. Leaf defenses involve phenylpropanoid metabolism genes, transcription factors, and hormone signaling modules that coordinate local and systemic responses, while root chitinase genes directly degrade fungal cell walls and block infection in a strain-specific and time-dependent manner. Together, these modules enable sugar beet to effectively resist pathogens across tissues and varying infection pressures (Figure 1).

### 2.2. Viral Pathogenesis in Sugar Beet Caused by Different Viruses

Virus yellows (VY), one of the most economically significant viral diseases, is transmitted primarily by aphids (*Myzus persicae*) and poses a major threat to beet production. Major viruses, including beet mild yellowing virus (BMYV), beet chlorosis virus (BChV), beet yellows virus (BYV), and beet mosaic virus (BtMV), are responsible for yellowing in sugar beet [46], leading to yellowing and necrosis of the leaves, which reduces photosynthetic capacity and impacts sugar yield. Virus infection could result in up to 50% yield loss, depending on the virus strain, the timing of infection, and the number of aphids transmitting the virus [47].

To explore eIF-mediated recessive resistance breeding against BMYV and BChV, a study has reported that genome-edited sugar beets were generated by knocking out different eIFs and infected with BChV and BMYV. The results showed that Bv-eIF(iso)4EKO plants significantly reduced BChV accumulation, with infection rates dropping from 100% to 28.86%, while no significant effect on BMYV accumulation was observed [48]. Mechanistically, these viruses depended on the interaction between their genome-linked protein (VPg) and host eIF proteins to initiate translation. Disruption of this interaction through knockout or mutation of beet BveIF(iso)4E proteins inhibited viral protein synthesis and replication, ultimately conferring resistance to the target viruses. Another transcriptome analysis revealed that 14 genes were significantly upregulated in BMYV-resistant beet varieties, 7 of which were associated with endoplasmic reticulum (ER) protein processing, whereas susceptible varieties exhibited more differentially expressed genes along with downregulation of photosynthesis-related pathways [49] (Table 1). Thus, it is hypothesized that resistance genes may activate the ER-associated degradation (ERAD) pathway to eliminate virus-induced misfolded proteins, acting in concert with defense proteins, transcription factors, and secondary metabolism-related genes to suppress viral accumulation, mitigate symptoms, and confer partial resistance to BMYV.

In addition, rhizomania is a major disease of sugar beet worldwide, caused by beet necrotic yellow vein virus (BNYVV) [11,50]. Four resistance genes have been identified: *Rz1*, *Rz2*, *Rz3*, and *Rz4*, with *Rz1* and *Rz2* being the main resistance factors [51,52,53]. *Rz1* was effective only against BNYVV, while *Rz2* provided resistance to both BNYVV and beet soilborne mosaic virus (BSBMV), with tissue-specific expression of *Rz2* observed in the roots. The tobacco (*Nicotiana benthamiana*) experiment demonstrated that co-expression of *Rz2* and BNYVV RNA1-4 triggered a resistance response in the leaves, and that the BNYVV triple gene block protein 1 (TGB1) served as the avirulence determinant for Rz2 [54] (Table 1). Rz2 encoded a CC-NB-LRR protein, and its resistance operated via a gene-for-gene mechanism: it recognized TGB1 and induced a local hypersensitive response (HR)-type cell death, effectively halting viral replication and spread in the roots. This targeted defense allowed beet to achieve broad-spectrum resistance against multiple soil-borne viruses. While in *B. macrocarpa*, transcriptome analysis revealed that 261 DEGs between BNYVV-infected plants and control plants, including those encoding pathogenesis-related (PR) proteins, WRKY/AP2 transcription factors, and extensins [55]. BNYVV infection activated the beet’s biotic stress response pathways: PRs directly inhibited viral proliferation, while transcription factors regulated hormone signaling and the expression of defense-related genes. Meanwhile, the virus interfered with genes involved in primary metabolism, such as carbohydrate and amino acid metabolism, and the plant maintained a defensive balance by differentially regulating these genes, thereby effectively resisting viral infection. However, the precise mechanisms underlying its disease resistance are not yet fully understood.

In summary, beet defenses against viral diseases could be grouped into three groups: eukaryotic translation initiation factors (e.g., *Bv-eIF*), which suppress viral translation by disrupting the VPg–eIF interaction; resistance genes (*Rz1* and *Rz2*), which target specific viruses and display tissue-specific expression; and a broad set of DEGs involved in hormone signaling, protein processing, and metabolic homeostasis. Working together, these genes inhibit viral replication, restrict viral movement within the plant, and stabilize host defense networks, thereby forming a coordinated and multilayered antiviral system in beet (Figure 1).

### 2.3. Resistance of Sugar Beet to the Bacterial Pathogen Pseudomonas syringae

*Pseudomonas syringae* secretes effector proteins or toxins that damage plant cell structures, resulting in stunted growth, reduced fruit yield, and higher control costs [56]. *Pseudomonas* species enhance plant immunity by inducing systemic resistance (ISR), particularly through the delivery of effector proteins via their type III secretion system (T3SS) [57]. Transgenic sugar beet, transformed with SP/HrpZ protein via *Agrobacterium rhizogenes*-mediated root transformation, significantly enhanced resistance to BNYVV, with higher resistance observed in the hairy roots after inoculation [58]. Studies have also shown that *P. marginalis* ORh26 colonized the roots of beet, reducing lesion size and pathogen spread caused by *P. syringae* pv. *aptata* in the leaves, while also activating peroxidase and phenylalanine ammonia-lyase, and upregulating defense genes such as *BvNPR1* and *BvMYC2* [59]. This activation of *BvNPR1* and *BvMYC2* by ORh26 triggered SA and JA signaling pathways and mobilizes systemic defense genes. Simultaneously, it strengthened local defenses by regulating ROS through *BvPOD* and promoted phytoalexin synthesis via *BvPAL*, reducing lesion size and pathogen proliferation. Mutants lacking T3SS completely lost this protective effect, underscoring the crucial role of T3SS in ORh26-mediated ISR in beet.

In beet, research on gene resistance to *P. syringae* is limited, especially regarding the regulation factors involved. While in red beet, the function of *BvHP4b* has been reported: as a histidine phosphotransfer protein in the cytokinin signaling pathway, it not only enhanced resistance to *P. syringae* pv. tomato DC3000 (Pst DC3000), but also coordinated the cell cycle through interaction with BvCDC2 [60] (Table 1). *BvHP4b* activated SA signaling pathway, upregulating defense genes such as *BvNPR1*, *BvPR1*, and *BvPR4* to enhance both local and systemic immunity in red beet. Simultaneously, it regulated cell wall–related genes (e.g., *BvXTH33*, *BvCESA6*) and the cellulase gene *BvCEL1*, thereby strengthened resistance to Pst DC3000 while coordinating taproot development, effectively integrating disease resistance with growth.

In summary, beet defense genes against *P. syringae* could be grouped into three functional modules: the signal transduction module, which activated SA and JA pathways and upregulated key immune genes such as *BvNPR1*, *BvMYC2*, and *BvPR1* to initiate defense responses; the metabolic defense module, in which *BvPOD* and *BvPAL* regulated ROS and synthesize antimicrobial secondary metabolites, respectively; and the structural and functional coordination module, exemplified by *BvHP4b*, which enhanced disease resistance while supporting taproot development, thereby integrating growth and defense (Figure 1).

### 2.4. Nematode Pathogenesis of Beet Cyst Nematode Infection Caused by Heterodera schachtii

The beet cyst nematode (*Heterodera schachtii*, BCN), a major cause of significant yield losses in sugar beet, induces stunted growth, leaf wilting, and “beard-like root” symptoms in infected plants [61]. Cultivating resistant varieties like Nemakill, Evasion, and Nematop, developed by hybridizing cultivated sugar beet with *Patellifolia procumbens*, is the most effective and economical control strategy, with genomic regions related to BCN tolerance revealing the genetic basis of resistance [62,63]. Transcriptome analysis revealed that beet resistance to BCN depended on a sustained and coordinated defense response [61]. At 4 and 10 days post-infection, the resistant cultivar Nemakill (carrying the resistance gene *Hs1^pro-1^*) displayed distinct gene expression patterns compared with a susceptible cultivar. In Nemakill, genes encoding CYSTM domain proteins, chitinases, F-box proteins, phenylpropanoid pathway enzymes, and CASP-like proteins were consistently upregulated, contributing to cellular stress protection, pathogen cell wall degradation, protein ubiquitination, secondary metabolite–based defense, and tissue barrier reinforcement. In contrast, defense genes in the susceptible cultivar 7112*SB36 were transiently induced at early stages and later suppressed, accompanied by disrupted hormone-related gene expression [61]. Exogenous application of MeJA and ethephon enhanced defense responses. The core of resistance lay in *Hs1^pro-1^*, derived from the wild beet relative *P. procumbens*, which encodes a protein containing leucine-rich repeat (LRR) and transmembrane domains. Its high expression coincided with the upregulation of defense genes, serving as a key trigger of resistance (Table 1). The *Hs1^pro-1^*-mediated defense pathway, in concert with JA/ET signaling, impairs nematode feeding site development and lifecycle completion, whereas susceptible cultivars fail to mount an effective defense due to imbalanced gene expression and disrupted hormonal signaling [61].

Besides the first nematode resistance gene *Hs1^pro-1^* cloned from a wild beet/sugar beet translocation line (encoding an NBS–LRR protein), a new resistance gene, *Hs4*, has been reported, which was localized to a specific region of approximately 230 kb through comparison of the resistant line TR520 and the susceptible translocation line TR363 [64]. This gene encodes a rhomboid-like protease, which is predicted to localize to the endoplasmic reticulum membrane. During nematode infection, feeding tubes connected to ER were induced to facilitate nutrient uptake by the nematode. *Hs4*, due to the high substrate specificity of its rhomboid protease, may directly cleave nematode effector proteins or disrupt ER structure, resulting in degeneration of the feeding sites (syncytia), blocking nutrient acquisition, and inhibiting nematode development. Knockout of this gene completely abolishes resistance, whereas its overexpression in the susceptible line 093161 confers full resistance [64] (Table 1), consistently highlighting the pivotal role of *Hs4* in nematode resistance.

The root-knot nematode resistance gene *R6m-1*, derived from wild beet (*B*. *maritima*), conferred broad-spectrum resistance to six root-knot nematode (*Meloidogyne* spp.) species in sugar beet [65]. Its expression markedly suppressed gall formation in roots, thereby disrupting abnormal tissue proliferation following nematode infection. Moreover, the closely linked CAPS marker NEM06 was associated with the *R6m-1* locus, and the protein encoded by this region was highly homologous to PERIANTHA in *Arabidopsis thaliana* and NIP1 in *Solanum lycopersicum*, suggesting that *R6m-1* might have acted as a transcription factor to regulate downstream defense genes and trigger resistance responses [65] (Table 1). Nevertheless, the specific downstream pathway components and their mechanisms remained to be experimentally confirmed. Other unmentioned biological stresses in sugar beet have also been extensively discussed in related reviews [20,21], offering new insights for breeding resistant varieties through genetic engineering or traditional breeding methods.

**Table 1 plants-15-00012-t001:** Summary of potential resistance genes and their functions in the biotic stress responses of beets in this study.

Types	Diseases	Pathogenic Organisms	Potential Resistance/Tolerance Genes	Gene Functions	References
Fungal stress	Cercospora Leaf Spot	*C. beticola*	*BvPAL*/*BvC4H*/*BvAREB1*	BvPAL and BvC4H are involved in plant defense responses, BvAREB1 activates ABA signaling to suppress BvPAL expression	[39,44]
Fungal stress	Cercospora Leaf Spot	*C. beticola*	*BvMBF1c*	Pathogen-responsive expression peaking at 48 h post-inoculation	[41]
Fungal stress	Cercospora Leaf Spot	*C. beticola*	*6 BvbZIPs*/*BvPR1*	BvbZIPs are significantly differentially expressed, and the BvPR1 protein is closely related to systemic acquired resistance	[42]
Fungal stress	Cercospora Leaf Spot	*C. beticola*	ROS-related genes/jasmonic acid-mediated defense genes	The resistant variety uses a three-stage defense strategy: early reinforcement of the cell wall and ROS scavenging, mid-stage amplification of jasmonic acid defense, and late-stage sustained defense without growth inhibition	[44]
Fungal stress	Root rot (RR)	*Fusarium* spp.	*BvSP2*, *BvSE2*	BvSP2 hydrolyzes fungal chitin and responds rapidly in resistant lines but delayed in susceptible lines. BvSE2 responds to field strains and is strongly expressed in susceptible leaves.	[45]
Viral stress	Virus Yellow	BMYV/BChV	*Bv-eIF*	Bv-eIF(iso)4EKO plants reduced BChV accumulation, but had no significant effect on BMYV accumulation.	[48]
Viral stress	Rhizomania	BNYVV	*Rz1/Rz2/Rz3/Rz4*	Rz2 is key for pathogen recognition, providing resistance to both BNYVV and BSBMV, with tissue-specific expression in the roots	[51,52,53,54]
Viral stress	Rhizomania	BNYVV	*TGB1*	TGB1 is the resistance determinant for Rz2	[54]
Viral stress	Rhizomania	BNYVV	*261 DEGs*	Enriched in response to biotic stimuli, primary metabolic processes, oxidoreductase activity, and hydrolase activity	[55]
Bacterial stress	Disease caused by *P. syringae*	*P. syringae*	*BvHP4b*	Acts as a positive regulator to enhance resistance to Pst DC3000	[60]
Nematode stress	BCN infection	*H. schachtii*	Defense-related genes	Consistently upregulated in resistant varieties	[61]
Nematode stress	BCN infection	*H. schachtii*	*Hs1^pro-1^*	Resistance to BCN	[61,63]
Nematode stress	BCN infection	*H. schachtii*	*Hs4*	Protect the plant from BCN infection	[64]
Nematode stress	Root-knot disease	*Meloidogyne* spp.	*R6m-1*	Confers broad-spectrum resistance, suppresses root gall formation	[65]

In summary, beet nematode resistance genes could be categorized into two functional modules: specific resistance genes (*Hs1^pro-1^*, *Hs4*, *R6m-1*), which directly block nematode infection or suppress root gall formation, and defense-supporting factors (chitinases, F-box proteins, phenylpropanoid pathway factors, and CASP-like proteins), which enhance cellular protection, degrade pathogen cell walls, and strengthen tissue barriers. Together, these factors act synergistically to establish the coordinated resistance system against nematodes in beet (Figure 1).

## 3. The Abiotic Stress Response of Beets

### 3.1. Tolerance of Sugar Beet to Drought and Waterlogging Stress

As a protective mechanism, under drought conditions, wild and fodder beet species exhibit increased leaf weight ratio and succulence index, along with thicker leaves [16,66], which help them better withstand heat and drought stress. Fodder beet is an important source of genes for drought tolerance, with higher genetic similarity to wild beet (*B. vulgaris* subsp. *maritima*), and its hybridization process is more convenient and efficient [16]. Heat shock proteins (HSPs) are essential in plants’ drought stress response, with *BvHSP70s* in beet tolerance variety KWS0143 showing differential regulation, as *BvHSP70-4*, *BvHSP70-13*, and *BvHSP70-14* were downregulated, while *BvHSP70-17* and *BvHSP70-20* were upregulated under drought stress [67], and these genes have not been functionally validated in previous studies. Through GWAS analysis of 328 beet germplasms, 11 drought-significant loci and 24 drought-related candidate genes were identified. Among these, 13 genes showed significant differential expression in extreme drought conditions, primarily associated with sugar metabolism, environmental stability, and photosystem repair [68], offering valuable insights for drought tolerance research in beet. For example, BVRB_7g160030 participated in starch biosynthesis and contributes to the regulation of carbohydrate metabolism and osmotic balance; BVRB_3g048910 facilitated phosphatidylcholine synthesis, thereby stabilizing cell membrane structure; and BVRB_7g166480 was involved in the assembly and repair of photosystem II (PSII) [68]. By modulating key physiological indicators such as soluble sugar and proline accumulation as well as SOD activity, these genes collectively enhance drought tolerance in beet seedlings. Proteomic analysis revealed that during drought stress recovery in beet, the responses of jasmonate-induced protein, salt-stress enhanced protein, and phosphatidylethanolamine-binding protein were delayed and persisted, potentially linking these proteins to the regulatory mechanisms underlying drought adaptation and memory effects [69]. Furthermore, many beet genes have been shown to perform specific functions when introduced into other species. Overexpressing the *BvHb2* gene in *A. thaliana* and tomato (*Solanum lycopersicum*) plants significantly enhanced their tolerance to drought and osmotic stress, helping them resist wilting induced by drought conditions [70] (Table 2).

In addition, waterlogging stress causes an imbalance in water distribution within plants and suppresses photosynthetic carbon assimilation in beet through both stomatal and non-stomatal limitations. The results of waterlogging treatment on the waterlogging-tolerant cultivar KUHN1260 (KU) and the sensitive cultivar SV1433 (SV) revealed that waterlogging stress significantly disrupted chloroplast ultrastructure, reduced photosynthetic pigment content and Rubisco activity, and downregulated key photosynthetic enzyme genes, including *BvRuBP*, *BvGAPDH* and *SvPRK* [71]. Concurrently, waterlogging induced ROS accumulation including H_2_O_2_ and O_2_^−^, resulting in oxidative damage. In contrast, the waterlogging-tolerant cultivar KU maintained higher activities of antioxidant enzymes—SOD, POD, CAT, and APX—along with elevated expression of their corresponding genes, efficiently scavenging ROS, mitigating membrane lipid peroxidation (as indicated by lower MDA levels), and preserving chloroplast integrity and photochemical efficiency [71]. These combined responses contribute to the cultivar’s markedly enhanced waterlogging tolerance (Table 2).

In summary, beet tolerance to drought and waterlogging stress depended on the coordinated action of multiple gene classes: heat shock proteins (BvHSP70s) assisted in maintaining protein homeostasis; sugar metabolism and osmotic regulation factors (e.g., BVRB_7g160030) maintained carbohydrate balance and osmotic stability; membrane structure and antioxidant factors (e.g., BVRB_3g048910) protected cell membranes and scavenged reactive oxygen species; and photosystem repair genes (e.g., *BvRuBP*) maintained photosynthesis and chloroplast stability. The synergistic action of these genes collectively enhanced beet stress adaptation and provided a basis for breeding drought- and waterlogging-tolerant varieties (Figure 2).

### 3.2. Tolerance of Beet to Salt and Alkali Stress

As a salt-tolerant plant, beet’s sensitivity to salt varies depending on the composition of salts in the water and the plant’s growth stage. To adapt to saline-alkali soils, sugar beet has developed a range of structural and physiological strategies, including regulating the distribution of salts and other solutes, and enhancing water retention [72,73,74]. Among these, betaine is a key osmotic regulator for salt-tolerant plants. Studies have shown that under normal conditions, betaine content is higher in young leaves and lower in old leaves, whereas under salt stress, betaine levels increase in all plant parts, particularly in the young leaves [75]. Additionally, under salt stress, the content and activity of chlorophyll, malondialdehyde (MDA), peroxidase (POD), superoxide dismutase (SOD), and catalase (CAT), also changed in beet [76,77].

The molecular mechanisms of beet under salt stress are primarily regulated by key genes, and recent studies have highlighted the crucial roles these genes play in adapting to high-salinity environments (Table 2). Beet adaptation to salt stress is not governed by a single gene, but is orchestrated by multilayered regulatory networks, including osmotic regulation, antioxidant defense, protein homeostasis, miRNA-mediated regulation, m^6^A modification, and other related regulatory pathways. Under salt stress, osmotic regulation and antioxidant defense systems serve as fundamental mechanisms for maintaining cellular homeostasis. For example, *BvM14-SAMS2* encoded S-adenosylmethionine synthase (SAMS) and enhanced tolerance to salt and H_2_O_2_ stress by regulating polyamine metabolism and the antioxidant system [78]. Under salt stress, *BvM14-SAMS2* was strongly induced in roots (12 h) and leaves (24 h), catalyzing the synthesis of SAM. SAM accumulation promoted polyamine metabolism, thereby protecting membrane integrity; concurrently, the activities of SOD, CAT, and POD were elevated to scavenge ROS, while proline content increased to maintain osmotic balance and metabolic homeostasis, collectively enhancing beet stress tolerance. In contrast, *AtSAMS3* knockout mutants were more sensitive to salt and H_2_O_2_, whereas complementation with *BvM14-SAMS2* restored tolerance (Table 1).

In addition, protein homeostasis regulation is one of the key mechanisms by which plants adapt to salt stress. Another gene, *BvM14-cystatin*, enhanced beet tolerance to salt stress by specifically inhibiting cysteine proteases, thereby reducing excessive protein degradation, maintaining nuclear function and membrane stability, and protecting key physiological processes such as photosynthesis [79]. *BvM14-cystatin* was constitutively expressed in roots, stems, leaves, and flowers (with higher levels in roots and stems), and its transcription was strongly induced by salt stress in beet, increasing approximately sixfold in leaves and about threefold in roots compared with controls. Transgenic *A. thaliana* plants expressing BvM14-cystatin exhibited alleviated root growth inhibition and improved survival under salt stress, further confirming its positive role in regulating plant responses to salinity (Table 1).

Beyond transcriptional regulation, miRNA-mediated post-transcriptional regulation plays a vital role in beet adaptation to salt stress. Beet seedlings displayed a “turgid-wilted-recovered” adaptive phenotype under salt stress. Integrative mRNA and miRNA sequencing analyses identified three key genes—*aldh2b7*, *thic*, and *δ-oat*—along with their regulatory miRNAs [80]. Under salt stress, miRNA-mediated repression of *aldh2b7* was alleviated, leading to its upregulation for detoxification of harmful aldehydes in coordination with secondary metabolism pathways. Similarly, release of miRNA inhibition on *δ-oat* enhanced proline synthesis to maintain osmotic balance while supporting amino acid homeostasis. Furthermore, miRNAs appeared to cooperate with hormone signaling to regulate *thic* expression, coordinating cofactor supply to maintain overall metabolic balance.

Recent studies have further shown that m^6^A-mediated epitranscriptomic modifications are also critical for modulating the expression of salt stress-responsive genes. *BvALKBH10B*, as an m^6^A demethylase involved in abiotic stress responses, was found to be upregulated in the leaves under salt stress. It targeted key salt-tolerance genes (such as *BvNHX1*, *BvCMO*, and *BvNXN2*) as well as genes related to phospholipid signaling, promoting their m^6^A demethylation and increasing their expression [81,82]. Through *BcNHX1* maintaining ion homeostasis, *BvCMO* enhancing the synthesis of osmotic regulators, and *BvNXN2* strengthening antioxidant protection—together with coordination of hormone regulation and protein ubiquitination—*BvALKBH10B* ultimately enhanced salt tolerance in beet (Table 2).

The analysis of gene families also revealed the responses of different members to salt stress, such as the MADS-box, SnRK2, bZIP, WD40, GRAS and HDA families [42,83,84,85,86,87], highlighting that multiple gene families cooperate through various mechanisms to regulate beet adaptation to salt stress. For instance, the bHLH gene family has also been reported to be involved in salt stress, with *BvbHLH93* expression significantly increasing treated with 200 mM NaCl, enhancing antioxidant enzyme activity by upregulating the expression of *SOD* and *POD* genes, while downregulating the expression of *RbohD* and *RbohF*, thereby reducing ROS production [88]. The salt inducibility of *bHLH137* and its bHLH motifs in the promoters of salinity-regulated genes suggested it may be a key factor in the salinity response [14].

Beet responds differently to neutral and alkaline salt stresses, with low salt concentrations promoting growth, while high neutral salt inhibits growth and photosynthesis more than alkaline salt [32]. Analysis of the beet cultivar H004 revealed that a low salt concentration (25 mM) increased gibberellin (GA_1_₊_3_) levels and upregulated genes involved in cuticle, cork, wax, and sesquiterpene/triterpene biosynthesis (e.g., FAR, squalene monooxygenase), thereby promoting beet growth. While at a high concentration (100 mM), beet showed greater tolerance to alkaline salts, which maintained homeostasis through the specific activation of monoterpene biosynthesis, amino acid metabolism, and starch–sucrose metabolism genes. In contrast, high neutral salt markedly decreased IAA (−18%) and increased ABA (+46.3%), induced DNA replication genes (e.g., MCM family), suppressed cuticle, cork, wax and linoleic acid metabolism genes, and regulated osmotic balance via soluble sugar accumulation, ultimately causing more pronounced growth inhibition under high neutral salt [89]. Moreover, analysis of the beet variety KWS0143 under short-term and long-term alkaline stress revealed that, at the physiological level, alkaline stress inhibited growth and reduced photosynthetic parameters (such as net photosynthetic rate and stomatal conductance), while proline accumulation and POD activity increased, alleviating oxidative damage. While at the transcriptomic level, 975 and 383 DEGs were identified under short-term and long-term stress, respectively, with 24 differential alternative splicing (DAS) genes also involved in regulation, including genes such as glutamyl-tRNA reductase 1, ethylene-insensitive protein 2, and metal tolerance protein 11 [90]. These associated metabolic and gene changes collectively helped the beet cope with damage caused by high pH and ionic stress (Table 2). Studies have also shown that under 50 mM alkaline conditions, the transcription levels of BvWRKY10 and BvWRKY16 in the roots and shoots of beet seedlings were significantly higher compared to the control conditions [91]. BvWRKY10 exhibited an expression pattern in the aerial parts characterized by “induction at low concentration, peaking at moderate concentration (50 mM NaHCO_3_), and maintaining at high concentration,” while BvWRKY16 reached its peak expression in roots at 50 mM NaHCO_3_, demonstrating clear tissue specificity and concentration dependence. Both genes activated downstream ion homeostasis genes (e.g., BvNHX, BvHKT), antioxidant enzymes (e.g., SOD, POD), and osmotic regulation-related factors, which together alleviated photosynthetic damage in leaves and maintained ion uptake in roots, thereby sustaining cellular homeostasis and enhancing beet tolerance to alkaline stress (Table 2). Other factors such as miRNA and non-coding RNAs also play roles under salt stress, which could enhance the salt tolerance and yield of beet [92].

Overall, the adaptation of beet to salt stress relied on the coordinated action of multiple functional factors. Osmotic balance-related genes (*δ-oat*, *BvCMO*) maintained cellular homeostasis by enhancing the synthesis of osmolytes such as proline and betaine; antioxidant and transcriptional regulatory genes (*BvM14-SAMS2*, *BvbHLH93*, *BvNXN2*) enhanced ROS scavenging by increasing SOD and POD activities or repressing ROS-producing genes; structural and metabolic genes (*BvM14-cystatin*) prevented excessive protein degradation and removed harmful aldehydes; and epigenetic and transcriptional regulators (*BvALKBH10B*) precisely modulated the expression of downstream salt-tolerance genes through epigenetic modifications. In response to alkaline stress, beet employed a multilayered gene regulation strategy to maintain physiological homeostasis: hormones (GA, ABA, and IAA) regulated growth; structural defense genes (FAR, squalene monooxygenase, and linoleic acid metabolism–related genes) reinforced the cuticle and wax barriers; metabolic homeostasis–related genes (mono-/sesquiterpene biosynthesis, amino acid, and starch-sucrose metabolism) maintained osmotic balance; and signal transduction and transcriptional regulatory genes (*BvWRKY10*, *BvWRKY16*, miRNAs/non-coding RNAs) controlled antioxidant responses and downstream gene expression (Figure 2).

### 3.3. Tolerance of Beet to Extreme Temperature Stress

Heat stress is a major environmental factor affecting the growth and productivity of beet, and different varieties exhibit varying responses to heat stress. Analysis of the expression of 334 *Hsp* genes in drought-resistance (DR), drought-sensitive (DS), and wild-type beet (*B. maritima*) revealed that the DR variety upregulated *BvHsp70-22*, *BvHsp90-03*, *BvHsp40-03*, and *BvHsp60-28* under heat stress (50 °C), with *BvHsp40-03* being crucial for heat tolerance [93]. DR beet responded to drought stress by significantly upregulating *BvHsp70-22* and *BvHsp90-03*: the former maintained protein folding and prevented aggregation through ATP-driven chaperone activity, while the latter regulated signaling protein activity. Together, they coordinated protein homeostasis and indirectly reduced ROS production, thereby enhancing drought tolerance. In contrast, DS primarily relied on *BvHsp60-01*, which only assisted in basic protein assembly within organelles; due to the lack of broader network regulation and low stress-response efficiency, their tolerance was limited. Wild-type beet highly expressed *BvsHsp-34/38*, which rapidly bound denatured proteins via α-crystallin domains to form protective complexes, reducing damage accumulation, while also coordinating osmotic adjustment to effectively mitigate the cellular impacts of abiotic stress [93]. In addition, under heat stress (40 °C), the beet small ubiquitin-like modifier (SUMO) system played a pivotal role through a “prioritize energy conservation—then activate repair” strategy. During the early stage, most SUMO components were suppressed to reduce energy consumption and help the plant cope with stress. By 12 h, *BvSIZ1* was significantly upregulated, precisely modifying heat shock proteins (e.g., BvHSP70s) and ribosomal proteins, with the former enhancing protein folding and the latter maintaining translation stability. At the same time, *BvESD4* was upregulated, dynamically regulating the balance between SUMOylation and deSUMOylation to ensure efficient and accurate modifications [94]. Overall, the SUMO system mitigated protein denaturation and cellular structural damage, thereby significantly enhancing beet heat tolerance (Table 2).

Low temperature stress, as another challenge for beet growth, particularly affects seedlings when exposed to freezing temperatures (below 0 °C) during early development, severely impacting seedling survival rates and limiting sucrose yield in mature plants [95]. Since vernalization requires prolonged exposure to cold temperatures, beet, as a biennial plant, must have cold tolerance to survive the winter [96], making cold tolerance a core breeding goal for beet. One gene with cold tolerance potential, the novel aquaporin *BvCOLD1*, could enhance cold tolerance, resistance to various abiotic stresses, and tolerance to boron deficiency at different developmental stages when overexpressed in *A. thaliana* [97]. After 28 days at 10 °C, the majority of transgenic plants survived and continued to grow, with biomass increasing under water and salt stress but slightly decreasing under cold stress [97]. The *raffinose synthase 1* (*BvRS1*), *raffinose synthase 2* (*BvRS2*), and *galactinol synthase 1* (*BvGolS1*) genes isolated from beet showed distinct expression patterns after treatment at 4 °C. The expression of *BvRS1* increased rapidly, indicating it may play a key regulatory role in response to cold stress. In contrast, the response of *BvRS2* was slower, suggesting its involvement in long-term adaptation. The higher expression of *BvGolS1* in the roots may help enhance cold tolerance in beet by promoting the accumulation of galactinol [98]. In addition, under low temperatures, another important factor was the vacuolar myo-inositol transporter BvINT1;1, which was markedly upregulated and transported vacuolar myo-inositol into the cytosol via an H^+^-coupled mechanism to provide substrates for raffinose synthesis [99] (Table 2). Raffinose served both as an osmoprotectant to reduce ice formation and as a direct scavenger of ROS to maintain redox homeostasis. At the same time, BvINT1;1-mediated myo-inositol transport indirectly modulated the activity of antioxidant enzymes (e.g., SOD, CAT). In BvINT1;1 loss-of-function mutants (*bvint1;1*), insufficient cytosolic myo-inositol limited raffinose synthesis to ~35% of wild-type levels, resulting in ROS accumulation, decreased photosystem II efficiency, and increased non-regulated energy dissipation, ultimately reducing taproot biomass and significantly impairing cold tolerance [99].

Transcription factor family analysis revealed that the BvSPLs in beet contribute to tolerance against low temperature and various abiotic stresses through core members and their coordinated network [100]. The key gene *BvSPL3* was rapidly upregulated under low-temperature stress, activating leaf genes involved in photosynthetic protection and ROS scavenging, while its high expression in mature taproots likely regulates parenchyma cell division and sucrose transporter genes (e.g., *BvTST2.1*), promoting taproot enlargement and sugar accumulation. *BvSPL6*, a root-specific stress-responsive gene, induced the synthesis of osmotic regulators (soluble sugars and proline) and activated stress signaling pathways under multiple abiotic stresses, acting in concert with *BvCPD* and other genes to support vascular bundle development. Among other *BvSPL* genes, *BvSPL5/7* (products of segmental duplication) respond dynamically to short-term stress, while *BvSPL2/8* indirectly modulate the expression of family members through proteins such as XP_010673830.1 (Table 2).

Studies have also shown that cold stress promotes beet’s adaptation to environmental changes by significantly upregulating genes in the ROS1 pathway (e.g., *BvROS1*, *BvNPX1*, *BvPIE1*), thereby promoting active DNA demethylation, while downregulating key genes in the RdDM pathway (e.g., *BvNRPD1*, *BvCLSY1*), suppressing maintenance and de novo DNA methylation [101]. These epigenetic modifications did not directly control gene expression; rather, they regulated the transcription of DNA (de)methylation-related genes, priming beet for subsequent transcriptional reprogramming in response to environmental changes. Moreover, pre-exposing plants to moderate stress could enhance their tolerance to subsequent stresses. For example, in the KWS1176 cultivar, under low temperatures (1–4 °C), salt stress significantly upregulated the expression of genes related to abscisic acid, gibberellin signaling, and 3-ketoacyl-CoA synthase [102]. On one hand, ROS metabolism-related genes (*BvGST23*, *BvPOX42*) and antioxidant enzymes (SOD, POX, CAT) were activated to reduce the accumulation of O_2_^−^, and H_2_O_2_, alleviating oxidative damage. On the other hand, carbohydrate metabolism genes (*BvHXK*) were upregulated to promote sucrose breakdown, energy supply, and proline accumulation. Simultaneously, ABA and gibberellin signaling genes (*BvPP2C*, *BvPYL*, *BvDELLA*, *BvGID2*) and fatty acid elongation genes (*BvKCS1*, *BvKCS10*) were regulated to optimize photosynthetic efficiency, membrane stability, and stress signaling [102], thereby significantly mitigating low-temperature-induced growth inhibition and physiological damage in beet.

In summary, beet’s heat stress tolerance was mediated by BvHsp family members and the SUMO system, which maintain protein folding, repair damaged proteins, and stabilize translation. Cold stress tolerance, in contrast, relied on modules including *BvRS1/2*, *BvGolS1*, *BvCOLD1*, *BvINT1;1*, and the BvSPL family, which supply metabolic substrates, accumulate osmoprotectants, and integrate stress-response signals to mitigate ROS-induced damage, thereby enhancing beet’s resilience to extreme temperatures (Table 2 and Figure 2).

### 3.4. Tolerance of Beet to Heavy Metal Stress

Heavy metal-contaminated soils pose environmental stress, and studying beet’s tolerance to heavy metals like cadmium (Cd) and nickel (Ni) helps evaluate its adaptability and potential as a phytoremediation tool. Previous studies have shown that beet was tolerant to Cd and Ni contamination, withstanding up to 225.8 mg/kg of bioavailable Cd, but when Ni concentrations exceeded 75.4 mg/kg, it became lethal to the plant’s growth [17]. Beet primarily accumulated Cd and Ni in the above-ground parts and demonstrated a higher ability to absorb nickel under combined Cd and Ni contamination [17]. Cd exposure increased oxidative damage in sugar beet, suppressed antioxidant enzyme activity, and caused growth retardation [103]. Under Cd stress, Cd markedly induced the root expression of *BvHMA3* and *BvNRAMP3*, enhancing Cd uptake and internal accumulation and thereby disrupting cellular structures. At the same time, it downregulated *BvIRT1* and reduced root Fe-chelate reductase (FCR) activity, inhibiting Fe uptake and ultimately leading to Fe deficiency and decreased photosynthetic efficiency [103] (Table 2). Although beet has been reported to absorb zinc (Zn), its efficiency in Zn phytoremediation remained relatively low. The metabolic responses of beet roots vary under different zinc (Zn) concentrations, with low and mild excess Zn primarily coping with toxicity by enhancing glycolysis [104]. While high concentrations of Zn significantly inhibited beet growth, causing root damage and symptoms of iron deficiency in the leaves, which resulted in reduced chlorophyll and carotenoid levels and decreased photosystem Ⅱ efficiency [105]. In addition, overexpression of the *Streptococcus thermophilus StGCS-GS* gene in beet significantly enhanced its tolerance to Cd, Zn, and copper (Cu), along with improved metal accumulation capacity [106] (Table 2), highlighting its strong potential for phytoremediation of multiple heavy metal contaminations.

**Table 2 plants-15-00012-t002:** Summary of potential resistance genes and their functions in the abiotic stress responses of beets in this study.

Types	Potential Resistance/Tolerance Genes	Gene Functions	References
Drought stress	*BvHSP70*	BvHSP70-4, BvHSP70-13, BvHSP70-14 downregulated, BvHSP70-17, BvHSP70-20 upregulated	[67]
Drought stress	*BvHb2*	Enhances drought and osmotic stress tolerance when overexpressed in *A. thaliana* and tomato	[70]
Waterlogging stress	*SvRuBP*, *SvGAPDH*, *SvPRK*	Expression levels significantly reduced under waterlogging stress	[71]
Salt stress	*BvM14-SAMS2*	Upregulated under salt stress, could enhance salt tolerance and antioxidant system	[78]
Salt stress	*BvM14-cystatin*	Overexpression in *A. thaliana* enhances salt tolerance	[79]
Salt stress	*BvbHLH93*	Upregulated under salt stress (200 mM NaCl) and enhances antioxidant enzyme activity	[88]
Salt stress	*BvbHLH137*	Regulates salt stress response through its bHLH motif in the promoters of salt stress-responsive genes	[14]
Salt stress	*BvALKBH10B*	Upregulated under salt stress	[81,82]
Alkaline stress	*Glutamyl-tRNA reductase 1*, *ethylene-insensitive protein 2*, and *metal tolerance protein 11*	DEGs identified under alkaline stress	[90]
Alkaline stress	*BvWRKY10, BvWRKY16*	Upregulated under alkaline stress	[91]
Heat stress	*BvHsp70-22*, *BvHsp90-03*, *BvHsp40-03*, *BvHsp60-28*	Upregulated under heat stress (50 °C)	[93]
Heat stress	*BvSIZ1*, *BvSAE2*	Significantly induced under high-temperature stress (40 °C)	[94]
Cold stress	*BvCOLD1*	Enhances cold tolerance and abiotic stress resistance when overexpressed in *A. thaliana*	[97]
Cold stress	*BvRS1*, *BvRS2*, *BvGolS1*	BvRS1 rapidly upregulates, BvRS2 responds more slowly, and BvGolS1 shows higher expression in the roots at 4 °C	[98]
Cold stress	*BvSPL6*, *BvSPL3*, *BvSPL2*, *BvSPL5*	BvSPL6 is upregulated in roots, BvSPL3 in leaves, and BvSPL2 fluctuates, while BvSPL5 decreases under −4 °C cold stress	[100]
Cold stress	*BvINT1;1*	Vacuolar myo-inositol transporter, crucial for maintaining cellular redox balance and cold tolerance	[99]
Heavy metal stress	*BvIRT1*, *BvHMA3*, *BvNRAMP3*	BvIRT1 is downregulated, while BvHMA3 and BvNRAMP3 are upregulated under Cd stress	[103]
Heavy metal stress	*StGCS-GS*	Overexpressing StGCS-GS in beet boosted its tolerance to Cd, Zn, and Cu stress	[106]

Together, beet’s tolerance to heavy metals such as Cd, Ni, and Zn primarily relies on multiple mechanisms. First, the metal uptake and transport module (e.g., *BvHMA3*, *BvNRAMP3*) could facilitate Cd absorption and internal accumulation. Then the metabolism and detoxification module (e.g., *StGCS-GS*) could enhance beet’s tolerance and accumulation capacity for multiple metals. Although some key genes provide critical support for beet’s adaptation to heavy metal stress, current knowledge is still limited, particularly regarding the molecular mechanisms of Cd tolerance, preferential uptake under Cd-Ni co-contamination, and differential responses to Zn stress, highlighting the need for further research using omics and gene-editing approaches (Figure 2).

## 4. Conclusions and Future Prospects

Beet, as a crucial economic crop, faces a variety of abiotic and biotic stress challenges that significantly impact its growth, development, and yield. Beet germplasm resources encompass local varieties, breeding lines, and wild types of *B. vulgaris* and its wild relatives (e.g., *B. maritima*), harboring valuable genes for stress tolerance, disease resistance, and quality traits, and representing key resources to broaden the narrow genetic base of beet [5,6,107]. Worldwide, several specialized beet germplasm databases have been established, including the International Database for Beta from Julius Kühn-Institute (JKI)—Federal Centre for Cultivated Plants (https://www.julius-kuehn.de/en/), the *Beta vulgaris* Resource (https://bvseq.boku.ac.at/), and the BeetBase (https://beetbase.scinet.usda.gov/), providing efficient platforms for the exploration, sharing, and introgression of resistance genes.

Interspecific hybridization between beet (*B. vulgaris*) and its wild relatives (e.g., *B. corolliflora*, *B. maritima*, or *B. patellaris*) for the introgression of beneficial genes is a key strategy to broaden the genetic base and enhance resistance to diseases, pests, and environmental stresses. Molecular cytogenetic and marker-assisted techniques provide precise support for this process: genomic in situ hybridization (GISH) could distinguish alien from cultivated chromosomes in the hybrids, fluorescence in situ hybridization (FISH) enables precise localization of target genes, and amplified fragment length polymorphism (AFLP) facilitates the assessment of genetic diversity following gene introgression [108,109,110]. For example, in the monosomic addition line M14, GISH directly detected the transmission of chromosomes from the parental *B. corolliflora*, and two M14-specific clones obtained from the chromosome 9 BAC microarray were localized near the telomeric region of the long arm, with the corresponding sequences exhibiting hemizygosity [108].

Building on existing research and in light of advances in high-throughput omics and precise gene-editing technologies, beet breeding and physiological studies continue to benefit from these innovations. Transcriptomics uses high-throughput analyses to precisely identify core factors involved in beet’s responses to salt, drought, and other abiotic/biotic stresses, while metabolomics focuses on key quality components such as betalains and sugars, revealing their dynamic changes across different developmental stages and stress conditions [111,112]. By integrating transcriptomic and metabolomic data, a “phenotype-metabolite-factor” regulatory network could be constructed [113,114,115], offering a solid theoretical foundation for beet genetic improvement and industrial advancement. CRISPR/Cas9 and other gene-editing tools are also driving beet research into a new era of precise, design-based breeding [21,116]. Their applications are expected to expand from single-gene functional validation to the development of novel germplasm with practical breeding potential, shifting the focus from improving individual traits to the coordinated enhancement of multiple traits, including stress tolerance, quality, and yield [117]. However, low transformation efficiency, complex and genotype-dependent regeneration systems, high ploidy and heterozygosity, and off-target risks remain major technical challenges that constrain their broad application in beet.

In this study, we summarize changes in beet under different stress conditions, focusing on the roles of key genes associated with stress tolerance (Figure 1 and Figure 2). As research advances, more disease-resistant genes in beet are being identified and functionally validated, strengthening immune responses through the expression of specific resistance genes (Table 1). Furthermore, in response to abiotic stresses such as drought, salinity, and low temperatures, beet could enhance its tolerance by regulating the expression of genes involved in metabolic pathways, hormone signaling, and antioxidant defense mechanisms (Table 2). Studying the genes involved in disease resistance and stress tolerance in beet, alongside modern breeding technologies, offers valuable theoretical and practical insights for its improvement. The application of CRISPR/Cas9 technology, in particular, enables precise genome editing to enhance stress tolerance, paving the way for new possibilities in beet breeding [21].

Future research may focus on the stress tolerance mechanisms of beet under various environmental conditions, particularly by enhancing pathogen resistance, as well as drought and salt-alkali tolerance, through overexpression and gene editing technologies (such as CRISPR/Cas9). By identifying resistance genes from wild relatives or stress-tolerant varieties and integrating multi-omics technologies like transcriptomics, proteomics, and metabolomics, we could pinpoint key genes and regulatory networks involved in stress responses, advancing precise breeding efforts. Based on these findings, a molecular marker database for stress tolerance could be established to support the selection and breeding of beet resistant varieties. Additionally, by combining phenotypic data with molecular markers, a comprehensive regulatory network could be constructed to further uncover gene interactions and signaling pathways, providing more refined strategies and directions for the genetic improvement of beet.

## Figures and Tables

**Figure 1 plants-15-00012-f001:**
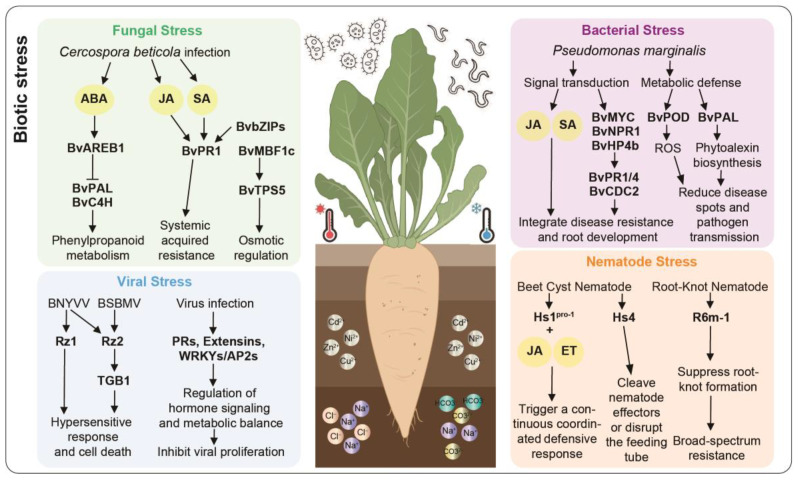
Overview of biotic stress responses of beet in this study. This figure systematically illustrates the core molecular response mechanisms of beet to various biotic stresses (fungal, viral, bacterial, and nematode). Each stress type is associated with specific genes and proteins that help beet adapt and survive under challenging environmental conditions.

**Figure 2 plants-15-00012-f002:**
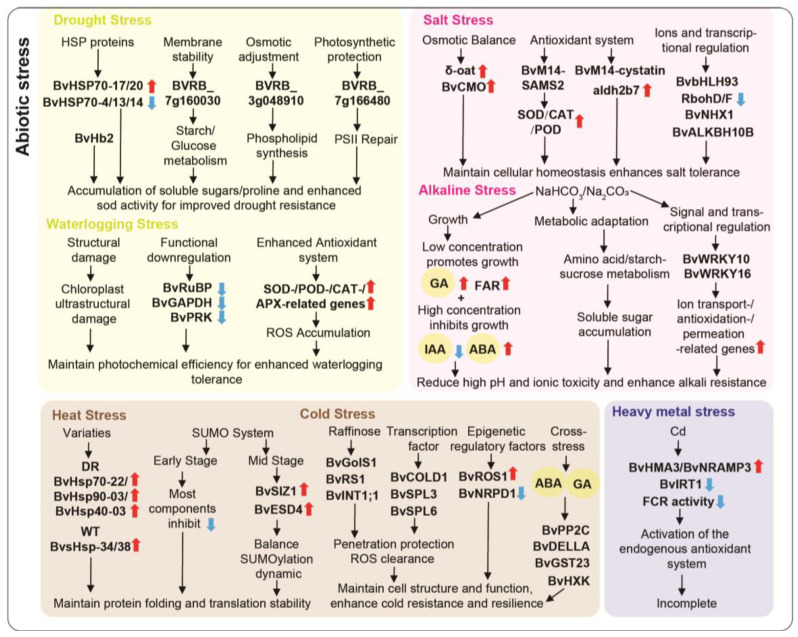
Overview of biotic stress responses of beet in this study. This figure systematically illustrates the core molecular response mechanisms of beet to various abiotic stresses (drought, salt, alkaline, waterlogging, heat, cold, and heavy metal). Each stress type is associated with specific genes and proteins that help beet adapt and survive under challenging environmental conditions. Note: Red arrow indicates up-regulation of gene/protein expression; blue arrow indicates down-regulation.

## Data Availability

No new data were created or analyzed in this study. Data sharing is not applicable to this article.

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
