# Peer review of "Advances in Beet (Beta vulgaris L.) Stress Adaptation: Focus on Transcription Factors and Major Stress-Related Genes"

_plants, 2025, doi:10.3390/plants15010012_

Round 1
Reviewer 1 Report
Comments and Suggestions for Authors
This review provides a solid and systematic summary of key genes and molecular mechanisms underlying stress tolerance in beet, serving as a useful reference for researchers in the field. Here are some suggestions:
- The abstract and introduction sections focus on stating which genes have been discovered by research, but they lack sufficient emphasis on the holistic and systemic view of how these genes constitute the core regulatory network of beet stress response. Suggestion: At the end of both the abstract and the introduction, clearly state that this review aims to integrate this scattered genetic information and reveal the key pathways behind it (such as ABA signaling, ROS scavenging, osmotic adjustment) and their interaction networks.
- Many parts of the text only report the expression changes of genes without deeply explaining the specific role and function of these changes in plant stress physiology. For example, the sentence "Under salt stress, the expression of *BvM14-SAMS2* gene was significantly upregulated... playing a crucial role in enhancing salt tolerance..." Suggestion: After describing a gene, add a sentence explaining the physiological and biochemical mechanism of its action.
- It is necessary to distinguish which genes are findings from transcriptome data analysis and which are functional genes validated through genetic experiments. It would be clearer to discuss them separately.
- It is recommended to add a short, refined summary paragraph at the end of each sub-section. This paragraph should categorize the genes mentioned earlier into different functional modules and explain how they work together synergistically.
- Figure 1 is currently a simple list-based schematic, whose information content highly overlaps with the main text and fails to effectively illustrate the mechanistic network. Suggestion: Upgrade Figure 1 into an integrated signal pathway map. Use arrows, blocking lines, and other symbols to categorize key genes into different physiological and biochemical pathways (such as "Hormone Signaling," "ROS Metabolism," "Osmotic Adjustment," "Pathogen Recognition") and visually show how these pathways respond to different biotic and abiotic stresses.
- Throughout the entire article, gene names are written in standard (non-italic) font, which does not comply with the universal naming conventions in the fields of genetics and molecular biology. Please correct this throughout the manuscript.
Comments on the Quality of English Language
There are many grammatical errors, long sentence structures and nonstandard expressions of scientific terms in this article. Please revise it carefully.
Author Response
#Review 1:
This review provides a solid and systematic summary of key genes and molecular mechanisms underlying stress tolerance in beet, serving as a useful reference for researchers in the field. Here are some suggestions:
- The abstract and introduction sections focus on stating which genes have been discovered by research, but they lack sufficient emphasis on the holistic and systemic view of how these genes constitute the core regulatory network of beet stress response. Suggestion: At the end of both the abstract and the introduction, clearly state that this review aims to integrate this scattered genetic information and reveal the key pathways behind it (such as ABA signaling, ROS scavenging, osmotic adjustment) and their interaction networks.
Response 1:
Thank you for the valuable suggestions. We have clarified at the end of the abstract and introduction that this review aims to integrate these dispersed gene data and highlight their potential interactions and key pathways (line 14-23, 30-109).
- Many parts of the text only report the expression changes of genes without deeply explaining the specific role and function of these changes in plant stress physiology. For example, the sentence "Under salt stress, the expression of *BvM14-SAMS2* gene was significantly upregulated... playing a crucial role in enhancing salt tolerance..." Suggestion: After describing a gene, add a sentence explaining the physiological and biochemical mechanism of its action.
Response 2:
Thank you for your careful review and helpful comments. We acknowledge that this manuscript previously offered limited explanations of the physiological roles and mechanisms of certain genes in plant stress responses, partly because the original studies did not provide relevant functional information. In response to your comments, we have supplemented the key genes with descriptions of their physiological and biochemical functions, including brief explanations of their roles in stress responses, to more clearly illustrate the relationship between gene expression changes and stress-related phenotypes (line 114-173, 195-207, 249-264, 278-318,336-350, 360-372, 399-433, 466-477, 500-520, 537-579, 596-600).
- It is necessary to distinguish which genes are findings from transcriptome data analysis and which are functional genes validated through genetic experiments. It would be clearer to discuss them separately.
Response 3:
Thank you for your detailed and constructive comments. In the revised manuscript, we have categorized the genes into two groups and discussed them separately (line 151-163, 199-207, 220-227, 278-295, 468-475, 548-560), thus providing a clearer presentation of their potential roles in plant stress responses along with the supporting evidence.
- It is recommended to add a short, refined summary paragraph at the end of each sub-section. This paragraph should categorize the genes mentioned earlier into different functional modules and explain how they work together synergistically.
Response 4:
Thank you for your insightful suggestions. We have added brief summaries at the end of each subsection to enhance the conciseness of the information and the clarity of the structure.
- Figure 1 is currently a simple list-based schematic, whose information content highly overlaps with the main text and fails to effectively illustrate the mechanistic network. Suggestion: Upgrade Figure 1 into an integrated signal pathway map. Use arrows, blocking lines, and other symbols to categorize key genes into different physiological and biochemical pathways (such as "Hormone Signaling," "ROS Metabolism," "Osmotic Adjustment," "Pathogen Recognition") and visually show how these pathways respond to different biotic and abiotic stresses.
Response 5:
Thank you for your thoughtful suggestions. Given that the references do not yet provide detailed information on signaling pathway networks, we have revised Figure 1 and added brief explanations in this revised manuscript, using arrows and inhibitory lines to assign key genes to different physiological and biochemical pathways, thereby illustrating their potential roles (line 685-692).
- Throughout the entire article, gene names are written in standard (non-italic) font, which does not comply with the universal naming conventions in the fields of genetics and molecular biology. Please correct this throughout the manuscript.
Response 6:
Thank you for your constructive feedback. All gene names in the revised manuscript have been revised and standardized in accordance with commonly accepted conventions in genetics and molecular biology.
- There are many grammatical errors, long sentence structures and nonstandard expressions of scientific terms in this article. Please revise it carefully.
Response 7:
Thank you for your valuable comments. We have carefully revised the manuscript to correct grammatical errors and improve lengthy sentence structures, thereby enhancing the accuracy and readability of the text.
Reviewer 2 Report
Comments and Suggestions for Authors
In the present study, authors summarized the Advances in Key Genes and Regulatory Mechanisms in Beet Stress Resistance. The topic may be interesting, but there is the critical problem in the present manuscript.
The overall framework is reasonable, but the description in each subtitle is not clear enough, and some logic is confusing. I hope the author will carefully consult the literature, reorganize it and make an in-depth review.
For example, In Introduction, lines 51-71, “The activation of the AREB cis-element A2 promoter and the induced expression of BvAREB1, which activated the ABA signaling path-way, suggested that ABA accumulation and its signaling pathway were the primary causes of the suppression of BvPAL expression”. This experiment can prove that they are related?
Lines 88-119, in the fungal pathogenic mechanism of Cercospora leaf spot of diabetes mellitus, it is first necessary to find out which regulatory factors are involved in the defense of the disease, and then summarize these factors with examples.
Lines 210-213, “Heat shock proteins (HSPs) are essential in plants’ drought stress response, with BvHSP70s in beet showing differential regulation, as BvHSP70-4, BvHSP70-13, and BvHSP70-14 were downregulated, while BvHSP70-17 and BvHSP70-20 were upregulated under drought stress.” Is this the comparison between varieties or the stress responsive? If it is a stress responsive, is this variety drought-tolerant or sensitive? Didn't explain it clearly. Whether these genes have verified their functions in previous studies, and there is no related review.
Comments on the Quality of English LanguageNeed to be further improved
Author Response
#Review 2:
In the present study, authors summarized the Advances in Key Genes and Regulatory Mechanisms in Beet Stress Resistance. The topic may be interesting, but there is the critical problem in the present manuscript.
The overall framework is reasonable, but the description in each subtitle is not clear enough, and some logic is confusing. I hope the author will carefully consult the literature, reorganize it and make an in-depth review.
For example, In Introduction, lines 51-71, “The activation of the AREB cis-element A2 promoter and the induced expression of BvAREB1, which activated the ABA signaling path-way, suggested that ABA accumulation and its signaling pathway were the primary causes of the suppression of BvPAL expression”. This experiment can prove that they are related?
Response 1:
Thank you for your detailed and constructive comments. Based on the existing reference, we could observe a correlation between the activation of the AREB cis-element A2 promoter, the induced expression of BvAREB1, and activation of the ABA signaling pathway. In the revised manuscript, we have adjusted the relevant statements to clarify that these results suggest ABA accumulation and its signaling pathway may be involved in suppressing BvPAL expression, rather than directly establishing a causal relationship (line 127-140).
Lines 88-119, in the fungal pathogenic mechanism of Cercospora leaf spot of diabetes mellitus, it is first necessary to find out which regulatory factors are involved in the defense of the disease, and then summarize these factors with examples.
Response 2:
Thank you for your constructive feedback. We have added the key regulatory factors involved in defense against this disease and summarized and categorized these factors with specific examples (line 114-117).
Lines 210-213, “Heat shock proteins (HSPs) are essential in plants’ drought stress response, with BvHSP70s in beet showing differential regulation, as BvHSP70-4, BvHSP70-13, and BvHSP70-14 were downregulated, while BvHSP70-17 and BvHSP70-20 were upregulated under drought stress.” Is this the comparison between varieties or the stress responsive? If it is a stress responsive, is this variety drought-tolerant or sensitive? Didn't explain it clearly. Whether these genes have verified their functions in previous studies, and there is no related review.
Response 3:
Thank you for your thoughtful suggestions. The description of BvHSP70 gene expression in the manuscript pertains to comparisons between varieties, and these genes have not been functionally validated in previous studies. We have added corresponding clarifications in the revised manuscript to better explain the experimental context and the validation status of these genes (line 336-340).
Reviewer 3 Report
Comments and Suggestions for Authors
The manuscript reviews the molecular mechanisms linking beet stress responses. The authors compile information from various studies to describe physiological, biochemical, and molecular responses under stress conditions. However, the current version lacks focus in some sections, with overlapping content and limited critical analysis of recent literature.
1. I think its a good fit for a mini-review rather than considering it as comprehensive review.
2. The introduction should be re-organized including specific background on plant-environment interactions and stress types; overview of stress signaling and molecular responses; importance of secondary metabolism in stress tolerance; and knowledge gap and objective of the paper.
3. In most of the casses, sentences repeat similar ideas about how plants adapt to stress; consolidation is needed.
4. Recent omics-based insights (transcriptome, metabolome, and gene-editing tools like CRISPR) should be included to reflect current advances (e.g., Zhao et al., 2024, Plant Physiology; Singh et al., 2023, IJMS).
5. In section 'Biotic Stress Responses' no sufficient link between specific molecular pathways or key signaling molecules (e.g., salicylic acid, jasmonic acid, ethylene) have been established.
6. I think the mechanistic link between stress perception (e.g., PRRs, effector recognition) and activation of secondary metabolite biosynthesis is weakly developed.
7. Several statements are general and should be supported by recent references (2022–2025) to reflect current understanding.
8. In section Abiotic Stress Responses: the narrative repeats some points from the Introduction and Biotic sections (e.g., stress perception and adaptation). Condense repetitive parts.
9. Include more discussion of ROS signaling, antioxidant defense systems, and their interaction with secondary metabolite pathways (e.g., flavonoid biosynthesis under drought).
10. Also Integrate key transcriptional regulators (e.g., DREB, NAC, bZIP) and hormonal cross-talk (ABA, auxin) that influence secondary metabolism under abiotic stress.
Author Response
#Review 3:
The manuscript reviews the molecular mechanisms linking beet stress responses. The authors compile information from various studies to describe physiological, biochemical, and molecular responses under stress conditions. However, the current version lacks focus in some sections, with overlapping content and limited critical analysis of recent literature.
1. I think its a good fit for a mini-review rather than considering it as comprehensive review.
Response 1:
Thank you for your valuable comments. We acknowledge that some sections may have appeared relatively dense. While our aim was to provide a broad overview of key genes involved in various stresses, we have revised the manuscript accordingly. We have added some content and descriptions, and refined the structure to improve clarity and better highlight the main points.
- The introduction should be re-organized including specific background on plant-environment interactions and stress types; overview of stress signaling and molecular responses; importance of secondary metabolism in stress tolerance; and knowledge gap and objective of the paper.
Response 2:
Thank you for the valuable comments. While the introduction primarily focuses on research progress in beet, based on your suggestion, we have also added relevant background, an overview of stress signals and molecular responses, the role of secondary metabolism, and the current knowledge gaps along with the objectives of this study, in order to enhance the overall structure of the introduction (line 30-109).
- In most of the casses, sentences repeat similar ideas about how plants adapt to stress; consolidation is needed.
Response 3:
Thank you for your constructive feedback. We have consolidated and refined the repeated statements regarding plant stress adaptation to make the text more concise and coherent.
- Recent omics-based insights (transcriptome, metabolome, and gene-editing tools like CRISPR) should be included to reflect current advances (e.g., Zhao et al., 2024, Plant Physiology; Singh et al., 2023, IJMS).
Response 4:
Thank you for your thoughtful suggestions. Based on your suggestion, we have added recent advances in omics research to the manuscript (line 644-660).
- In section 'Biotic Stress Responses' no sufficient link between specific molecular pathways or key signaling molecules (e.g., salicylic acid, jasmonic acid, ethylene) have been established.
Response 5:
Thank you for the valuable comments. Although current studies do not provide evidence for interactions among these signaling molecules, we have revised the “Biotic Stress Responses” section to include additional descriptions that more clearly illustrate their roles (Section 2, 3 and Figure 1).
- I think the mechanistic link between stress perception (e.g., PRRs, effector recognition) and activation of secondary metabolite biosynthesis is weakly developed.
Response 6:
Thank you for your valuable comments. Our focus is primarily on key stress-resistance genes in sugar beet, and many studies have not addressed the mechanistic links between stress perception and secondary metabolite biosynthesis. Nevertheless, we have added descriptions of their potential connections based on your suggestion (line 195-207, 399-425, 450-456, 486-493,571-579).
- Several statements are general and should be supported by recent references (2022–2025) to reflect current understanding.
Response 7:
Thank you for your thoughtful suggestions. We have revised the relevant content and added new references to refine and expand these points. Additionally, references from 2020–2025 now account for 89% of the total citations.
- In section Abiotic Stress Responses: the narrative repeats some points from the Introduction and Biotic sections (e.g., stress perception and adaptation). Condense repetitive parts.
Response 8:
Thank you for your detailed comments. We have simplified the “Abiotic Stress Responses” section to make the structure more concise and the text clearer.
- Include more discussion ofROS signaling, antioxidant defense systems, and their interaction with secondary metabolite pathways (e.g., flavonoid biosynthesis under drought).
Response 9:
Thank you for your thoughtful suggestions. Based on current references in this manuscript, we have added a discussion section on ROS signaling, the antioxidant defense system, and their potential interactions with secondary metabolite pathways (Section 2, 3 and Figure 1).
- Also Integrate key transcriptional regulators (e.g., DREB, NAC, bZIP) and hormonal cross-talk (ABA, auxin) that influence secondary metabolism under abiotic stress.
Response 10:
Thank you for your constructive feedback. Based on your suggestion, we have summarized the key transcriptional regulators and their interactions with hormones that play important roles in secondary metabolism under abiotic stress in the discussion section (Section 2, 3 and Figure 1).
Reviewer 4 Report
Comments and Suggestions for Authors
Beet (fodder and sugar beets) is one of the world's most important agricultural crops. The mechanisms of beet tolerance to various biotic and abiotic factors are actively studied, including at the genetic level. However, such a comprehensive review article is currently unavailable. In it, the authors summarize the genes involved in the tolerance of various beet varieties (Beta vulgaris) to various stressors. The review is divided into two parts: "The biotic stress responses of beets" and "The abiotic stress responses of beets." A separate section is devoted to "Conclusion and future prospects." Tables of biotic and abiotic response genes in beet plants are provided, as well as a schematic figure "Overview of biotic and abiotic stress responses of beets in this study." The review also examines transgenic beet lines and establishes the role of individual genes in stress tolerance. In summary, this review article serves as a reference guide to stress tolerance genes in beets. The authors titled the manuscript “Advances in….” However, an analysis of the Reference List revealed that of the 99 sources, only 40% were articles from the last 5 years. Therefore, the authors should increase the number of cited articles from the last 5 years.
Perhaps the authors should expand their information on the sugar beet genetic base through the introgression of resistance genes from wild relatives, molecular cytogenetic methods (GISH and FISH hybridization - genome- and fluorescent in situ hybridization), and AFLP analysis.
It is known that the Federal Center for Plant Breeding Research in Braunschweig, Germany, maintains a gene bank and an international database on sugar beets, which contains information on more than 10,000 accessions from 20 banks in different countries. Carriers of the corresponding genes are registered in the journal "Crop Science" and assigned an individual number, which authors can access.
There is no information on the BvSP2 and BvSE2 genes, which provide resistance to biotic stress – protection against fungal infection; there is no information on the R6m-1 gene, which provides resistance to plant helminthiasis.
Authors should include the Latin name of the beet species in the manuscript title and keywords. Additionally, the Introduction should indicate the family Amaranthaceae, to which beet belongs.
Author Response
#Review 4:
Beet (fodder and sugar beets) is one of the world's most important agricultural crops. The mechanisms of beet tolerance to various biotic and abiotic factors are actively studied, including at the genetic level. However, such a comprehensive review article is currently unavailable. In it, the authors summarize the genes involved in the tolerance of various beet varieties (Beta vulgaris) to various stressors. The review is divided into two parts: "The biotic stress responses of beets" and "The abiotic stress responses of beets." A separate section is devoted to "Conclusion and future prospects." Tables of biotic and abiotic response genes in beet plants are provided, as well as a schematic figure "Overview of biotic and abiotic stress responses of beets in this study." The review also examines transgenic beet lines and establishes the role of individual genes in stress tolerance. In summary, this review article serves as a reference guide to stress tolerance genes in beets. The authors titled the manuscript “Advances in….” However, an analysis of the Reference List revealed that of the 99 sources, only 40% were articles from the last 5 years. Therefore, the authors should increase the number of cited articles from the last 5 years.
Response 1:
Thank you for your thoughtful suggestions. We have added more references from the past five years as suggested, to better reflect the recent advances in the field. Additionally, references from 2020–2025 now account for 89% of the total citations.
Perhaps the authors should expand their information on the sugar beet genetic base through the introgression of resistance genes from wild relatives, molecular cytogenetic methods (GISH and FISH hybridization - genome- and fluorescent in situ hybridization), and AFLP analysis.
Response 2:
Our primary focus is on key stress-resistance genes in beet, thus the introgression of resistance genes from wild relatives as well as molecular cytogenetic methods (such as GISH, FISH) and AFLP analyses are not the main emphasis. Nevertheless, we have added relevant information in appropriate sections as suggested to provide additional background (line 622-643).
It is known that the Federal Center for Plant Breeding Research in Braunschweig, Germany, maintains a gene bank and an international database on sugar beets, which contains information on more than 10,000 accessions from 20 banks in different countries. Carriers of the corresponding genes are registered in the journal "Crop Science" and assigned an individual number, which authors can access.
Response 3:
Thank you for your thoughtful suggestions. We have added the relevant information in the manuscript including the website for accessing materials carrying resistance genes (line 622-630).
There is no information on the BvSP2 and BvSE2 genes, which provide resistance to biotic stress – protection against fungal infection; there is no information on the R6m-1 gene, which provides resistance to plant helminthiasis.
Response 4:
Thank you for your valuable comments. We have added information on BvSP2, BvSE2, and R6m-1 genes in the manuscript, which are involved in fungal infection defense and nematode resistance, respectively (line 165-173, 309-318).
Authors should include the Latin name of the beet species in the manuscript title and keywords. Additionally, the Introduction should indicate the family Amaranthaceae, to which beet belongs.
Response 5:
Thank you for your constructive feedback. We have added the Latin name Beta vulgaris L. to this manuscript title and keywords, and clearly indicated in the Introduction that sugar beet belongs to the family Amaranthacea.
Round 2
Reviewer 2 Report
Comments and Suggestions for Authors
The author has seriously answered the questions raised. But there still has a major revisions in the manuscript are needed before considering publication. In the abiotic stress response of beets,personally, I still feel a little confused. For example, in 3.2. Tolerance of beet to salt and alkali stress, lines 398-434, the existing key regulatory genes of beet salt and alkali tolerance should be classified and described, so it seems to be organized? Here, SAMS2 suddenly appears. I don't know what regulation mechanism you want to describe. Is it osmotic regulation or oxidation protection?
Comments on the Quality of English LanguageNeed to be further improved
Author Response
The author has seriously answered the questions raised. But there still has a major revisions in the manuscript are needed before considering publication. In the abiotic stress response of beets,personally, I still feel a little confused. For example, in 3.2. Tolerance of beet to salt and alkali stress, lines 398-434, the existing key regulatory genes of beet salt and alkali tolerance should be classified and described, so it seems to be organized? Here, SAMS2 suddenly appears. I don't know what regulation mechanism you want to describe. Is it osmotic regulation or oxidation protection?
Response 1:
Thank you for the detailed comments. In response to your suggestions, we have revised Section 3.2. Tolerance of beet to salt and alkali stress. The key regulatory genes in this manuscript have now been systematically classified according to their functions into modules, including osmotic regulation and antioxidant defense, protein homeostasis regulation, miRNA-mediated post-transcriptional regulation, and m⁶A epitranscriptomic modification. Notably, BvM14-SAMS2 has been assigned to the osmotic regulation and antioxidant defense module (line 399-445).
Reviewer 3 Report
Comments and Suggestions for Authors
Accepted in its current form.
Author Response
Thank you very much for your kind acceptance and valuable comments.
Reviewer 4 Report
Comments and Suggestions for Authors
The authors made significant revisions to the manuscript, in accordance with the reviewers' comments and suggestions. The authors responded to my comments and questions by changing the manuscript title and figure, adding references for the last five years, and information on some stress genes. However, the authors forgot to add new references to the References section - as in the first version, it contains 99 references, while 117 are needed (L. 658).
Author Response
The authors made significant revisions to the manuscript, in accordance with the reviewers' comments and suggestions. The authors responded to my comments and questions by changing the manuscript title and figure, adding references for the last five years, and information on some stress genes. However, the authors forgot to add new references to the References section - as in the first version, it contains 99 references, while 117 are needed (L. 658).
Response 1:
Thank you very much for your valuable comments. We have systematically updated the reference list, incorporated all newly added references into the References section, and ensured consistency with the citations in the revised manuscript.
Round 3
Reviewer 2 Report
Comments and Suggestions for Authors
The author has carefully revised the full manuscript as required. Here is a question: Where is thic gene in Table 2 in line 434-436, and what is the reference?
Comments on the Quality of English LanguageNo
Author Response
The author has carefully revised the full manuscript as required. Here is a question: Where is thic gene in Table 2 in line 434-436, and what is the reference?
Response 1:Thank you for your question. The thic gene is mentioned in the main text in lines 427–435, where it is described accordingly:
“Beyond transcriptional regulation, miRNA-mediated post-transcriptional regulation plays a vital role in beet adaptation to salt stress. Beet seedlings displayed a “turgid-wilted-recovered” adaptive phenotype under salt stress. Integrative mRNA and miRNA sequencing analyses identified three key genes—aldh2b7, thic, and δ-oat—along with their regulatory miRNAs [80]. Under salt stress, miRNA-mediated repression of aldh2b7 was alleviated, leading to its upregulation for detoxification of harmful aldehydes in coordination with secondary metabolism pathways. Similarly, release of miRNA inhibition on δ-oat enhanced proline synthesis to maintain osmotic balance while supporting amino acid homeostasis. Furthermore, miRNAs appeared to cooperate with hormone signaling to regulate thic expression, coordinating cofactor supply to maintain overall metabolic balance. ”
Reference:
[80] Zhang, Z.; Wang, L.; Chen, W.; Fu, Z.; Zhao, S.; E, Y.; Zhang, H.; Zhang, B.; Sun, M.; Han, P.; et al. Integration of mRNA and miRNA analysis reveals the molecular mechanisms of sugar beet (Beta vulgaris L.) response to salt stress. Scientific Reports 2023, 13, doi:10.1038/s41598-023-49641-w.